Positive affect and age as predictors of exercise compliance

Garcia Danilo 1 2 danilo.garcia@euromail.se danilo.garcia@neuro.gu.se
Archer Trevor 2 3
1 Institute of Neuroscience and Physiology, Centre for Ethics, Law and Mental Health (CELAM), University of Gothenburg , Gothenburg , Sweden
2 Network for Empowerment and Well-Being , Sweden
3 Department of Psychology, University of Gothenburg , Gothenburg , Sweden
Abdullah Jafri
Electronic publication date: 2014 Dec 16
Publication date: 2014
Volume: 2
Electronic Location ID: e694
Received 2014 Aug 10; Accepted 2014 Nov 21
Copyright: © 2014 Garcia and Archer
Copyright year: 2014
Copyright holder: Garcia and Archer
License: This is an open access article distributed under the terms of the Creative Commons Attribution License, which permits unrestricted use, distribution, reproduction and adaptation in any medium and for any purpose provided that it is properly attributed. For attribution, the original author(s), title, publication source (PeerJ) and either DOI or URL of the article must be cited.
License URL: https://creativecommons.org/licenses/by/4.0/

Keywords: Exercise, Positive affect, Negative affect, Age, The Archer ratio

Funding: This study was supported by the Bliwa Stiftelsen. The funders had no role in study design, data collection and analysis, decision to publish, or preparation of the manuscript.

==============================
Physical exercise is linked to individuals whose affect profiles are invariably positive and it induces anti-apoptotic and anti-excitotoxic effects, buttressing blood–brain barrier intactness in both healthy individuals and those suffering from disorders accompanying overweight and obesity. In this regard, exercise offers a unique non-pharmacologic, non-invasive intervention that incorporates different regimes, whether dynamic or static, endurance, or resistance. In this brief report we present a self-reported study carried out on an adolescent and adult population (N = 280, 144 males and 136 females), which indicated that the propensity and compliance for exercise, measured as the “Archer ratio”, was predicted by a positive affect. This association is discussed from the perspective of health, well-being, affect dimensions, and age.

The introduction of exercise regimes has been found to be beneficial under both laboratory and clinical conditions; this observation is particularly evident in consideration of various health biomarkers. Any bodily activity that enhances or maintains physical fitness implies the involvement of regular and frequent exercise. Morris & Schoo (2004) have defined exercise as a planned, structured physical activity with the purpose of improving one or more aspects of physical fitness and functional capacity. Physical exercise influences cognitive, emotional, learning and neurophysiological domains, both directly and indirect, thereby rendering it essential that this noninvasive, non-pharmacological intervention ought to form a part of long term health programs for children and adolescents. (Archer, 2014). In juvenile and adolescent populations, physical exercise holds benefits in association with the health of bones, cardiovascular fitness, healthy blood lipid profiles, psychological well-being and is linked inversely to levels of adiposity and stress (Loprinzi et al., 2012).

The types of exercise relevant here have been characterized on the basis of type, intensity, frequency, and duration, with either endurance or resistance capacity as the training endpoint (Mougios, 2010). Endurance exercise develops one’s ability to exert oneself over long periods of physical activity, whereas resistance exercise implies exerting resistance to the force of muscular contraction and elastic or hydraulic resistance (Ormsbee et al., 2009). Physical exercise has been shown to manifest marked improvements both in function and biomarker integrity (e.g., Archer, 2011; Archer, 2013; Archer & Fredriksson, 2010; Archer & Kostrzewa, 2011; Archer et al., 2011; Fredriksson et al., 2011). Thus, the benefits of regular physical exercise as a health-ensuring necessity over age, gender, occupation and affective status cannot be overestimated (Garcia et al., 2012; Palomo et al., 2008).

Nevertheless, the prevailing situation among youth populations suggests that these benefits are largely unrealized. For instance, the 2011 Centers for Disease Control and Prevention Youth Risk Behavior Surveillance System (2011) found that only 28.7% of healthy high school pupils showed physical activity levels that reached the federal guideline of 60 min moderate-to-vigorous intensity physical exercise. Exercise compliance was even less among those presenting chronic disorders, such as type I diabetes (Lukacs et al., 2012; Maggio et al., 2010). Nevertheless, there is evidence that physical exercise positively influences academic performance and well-being in children and adolescents (Archer & Garcia, 2014), suggesting a strong indication of its benefits on cognition and affect. Exercise compliance presents advantages, not only regarding reduced stress, anxiety and depression, but also improved self-esteem and psychological well-being (Dunton et al., 2014; Garcia et al., 2012; Gaz & Smith, 2012; Melnyk, Hrabe & Szalacha, 2013).

The affective domain presents a frequently studied issue within physical exercise interventions, not least since the ability and capacity to interact with other individuals modulates the positive interaction with oneself, thereby placing a premium on high levels of positive affect by presenting the conditions for developing personal characteristics and delivering affective qualities (Heidor & Welch, 2010). Positive and negative affect have been shown to reflect stable emotional-temperamental dispositions (Watson & Clark, 1994; Tellegen, 1993). For instance, one of the most used instruments to measure affect is the Positive Affect and Negative Affect Schedule (Watson, Clark & Tellegen, 1988), which was developed with the idea that positive and negative affect represent two orthogonal independent dimensions (see also Watson & Tellegen, 1985). Moderately intense physical exercise induces higher levels of positive affect and lower levels of negative affect in both younger (mean 20 years) and older (mean 56 years) adults (Barnett, 2013). Even in older adults (65–95 years), both strength-based and aerobic-based physical activity improves skills and mobility with concurrent improvements in mood (Martins et al., 2011). In a study of both younger and older adults, 144 participants of ages ranging from 19 to 93 years, assigned to a moderate intensity exercise group (15 min moderate intensity cycling) or a control group (15 min rating neutral images), it was observed that exercise increased high-arousal positive affect and decreased low-arousal positive affect in comparison with controls (Hogan, Mata & Carstensen, 2013), as well as improving cognitive performance on a working memory task. In community-dwelling older adults (over 55 years) presenting mild cognitive impairment, a physical exercise intervention (SMART) was shown to improve psychological well-being, cognitive functioning and quality-of-life (Gates et al., 2014). Moreover, the benefits are not only at the cognitive level. For example, Östhus and colleagues (2012) studied cardiovascular fitness (VO2max) and endurance training in younger and older individuals. They found that long-term endurance exercise training exerted a protective effect upon muscle telomere length in older adults—VO2max being associated positively with telomere length. In this context, it is important to point out that telomeres, nucleoproteic complexes located at the ends of eukaryotic chromsomes composed by non-coding repetitive sequences (McEachern, Krauskopf & Blackburn, 2000), seem to function as a mitotic ‘clock’, progressively shortening, the triggering of DNA damage response and apoptosis (Blackburn, 2001). Finally, physical exercise programs for older adults and elderly have improved dietary habits, memory for events and materials, emotional balance and the enjoyment of cultural, intellectual, affective and social activities (Caprara et al., 2013). Finally, Solberg et al. (2014) obtained improvements in most measures of well-being after four months of endurance training.

The purpose of the present study was to ascertain if affectivity and age predict compliance with frequent and intensive physical exercise using self-reported data from a sample consisting of Swedish high-school students and individuals employed in a number of administrative and skilled labor occupations.

Method

Ethical statement

According to the law (2003: 460, §2) concerning the ethical research involving humans, we arrived at the conclusion that the design of the present study (e.g., all participants’ data were anonymous and will not be used for commercial or other non-scientific purposes) required only informed consent from the participants.

Participants and procedure

A total of 280 participants (144 males and 136 females) were included in the analysis. This sample included high school pupils, university students, and also white-collar workers from the private and public sector (age mean = 25.60 sd = 12.81). All participants were residents of Gothenburg, Sweden. Participants were made aware that the study was anonymous, voluntary and that it took 15 min to complete all the reports. First, the participants completed the background questionnaire and then a battery of instruments including one measure of affect. In the background questionnaire we included the measures for exercise behavior.

Measures

Exercise behavior

Besides collecting demographic data (e.g., age, gender), the background questionnaire included two items to measure the frequency (“How often do you exercise?”; 1 = never, 5 = 5 times/week or more) and the intensity (“Estimate the level of effort when you exercise”; 1 = non or very low, 10 = Very High) of exercise behavior.

Affect

The Positive Affect Negative Affect Schedule (Watson, Clark & Tellegen, 1988) allows participants to respond on a 5-point Likert scale to what extent (1 = very slightly, 5 = extremely) they generally experienced the 20 adjectives encompassing 10 positive affect and 10 negative affect words within the last few weeks. The positive affect subscale consists of adjectives such as “strong”, “proud” and “interested.” The negative affect subscale consists of adjectives such as “afraid”, “nervous” and “ashamed.” The Swedish version has been used in a wide range of studies over the last decade (e.g., Garcia & Erlandsson, 2011; Nima et al., 2013; Schütz, Archer & Garcia, 2013). In the present study Cronbach’s α for the positive affect subscale was .82 and for the negative affect subscale .86.

Statistical treatment

The answers to both exercise-items were first standardized (i.e., transformed to z-scores) in order to summarize them into a composite measure for exercise behavior, that is, The Archer Ratio (Garcia & Archer, 2014). A principal components analysis, with oblimin rotation, suggested that a single primary factor accounted for at least 70.94% of the variance, thus supporting the calculation of The Archer Ratio. Further regression analysis was conducted using age, gender, positive affect and negative affect as the predictors and The Archer Ratio as the outcome. A correlation analysis showed that the variables weren’t correlated higher than .30 (see Table 1), thus lower than what is suggested concerning multicolliniarity (see, for example Tabachnick & Fidell, 2007, 88). A total of 10 participants (5% of the total participants) had not answered some of the questions and were therefore discarded from the analysis. The Archer Ratio has shown its validity by predicting actual exercise compliance in a population at a training facility (Garcia & Archer, 2014), even when compared to larger, strong, and validated scales such as The Godin–Shephard Leisure-Time Physical Activity Questionnaire (Godin & Shephard, 1985).

Table 1 Correlation between age, positive affect, negative affect, and the Archer ratio.

	Age	Positive affect	Negative affect	The Archer ratio	
Age	–				
Positive affect	.27**	–			
Negative affect	−.30**	−.04	–		
The Archer ratio	−.14*	.24**	.03	–	
Notes.

* Correlation is significant at the 0.05 level (2-tailed).

** Correlation is significant at the 0.01 level (2-tailed).

Results and discussion

A significant model emerged (F(4, 268) = 7.81, p < .001). Table 2 provides information about regression coefficients for the predictor variables (i.e., age, gender, positive and negative affect) entered predicting The Archer Ratio. Age and positive affect were the only significant predictors of exercise behavior. Specifically, exercise behavior decreased with age but was positively associated to positive affect.

Figure 1 The association of positive affect in the prediction of long-term compliance in the propensity to perform physical exercise, i.e., “exercise behavior.”

Table 2 The unstandardized and standardized regression coefficients for the variables entered into the regression model as predictors of The Archer ratio (i.e., exercise behavior).

Predictor	B	SE B	β	t	p	
Age	−.05	.02	−.22	−3.22	.001	
Gender	24	.35	.04	.69	.494	
Positive affect	1.13	.23	.30	4.95	<.001	
Negative affect	−.11	.24	−.03	−.45	.656	

The present study indicates that exercise compliance is positively associated with positive affectivity and negatively with age (Fig. 1). Currently, this account underlines the importance of individuals’ basal levels of positive affect that mobilizes the compliance and propensity for exercise. Affective status, whether negative, e.g., in anorexia, or positive, e.g., as a health measure, is linked with behaviors that maintain dietary habits (Engel et al., 2013). Regular physical exercise induces fuel utilization, which mobilizes the energetic cost of storing excess nutrients during relapse and alterations in circulating nutrients that may modulate appetite, thereby attenuating the biological drive to regain weight, involving both central and peripheral aspects of energy homeostasis. This may explain, in part, the utility of regular activity in preventing weight regain after weight loss (Steig et al., 2011). It provides a ‘scaffolding effect’ that alleviates the effects of TBI (Archer, 2013) and symptoms and biomarkers of depression (Archer, Josefsson & Lindwall, 2014). Exercise, particularly when linked to dietary restriction, offers a cheap and practical non-pharmacological, noninvasive intervention that, if introduced proactively, will provide marked elements of prevention. It has been recommended that physical exercise be perceived and employed in a similar manner to pharmaceutical, psychotherapeutic, physiotherapeutic, and other biosocio-medical interventions involving the basic and continuing education and training of health care personnel and evaluation of processes to assess their needs and to prescribe and deliver them, to reimburse the services and programs related to it, and to fund research on its efficacy, applicability, feasibility, compliance and interactions and comparability with other preventive, therapeutic, and rehabilitative domains (Vuori et al., 2013). Autophagy appears to offer the processes that physical exercise generates with marked health benefits involving life-span expansion, protection against several disease that compromise brain function and clear benefits for metabolic and bioenergetic dynamics. Accumulated evidence has underlined the premise that brain neural, muscular, neuroimmune and other physiologic systems are subject to the principle of “Use it-or-lose-it” intrinsic to all motor activity and exercise.

The present results associating positive affect but not negative affect with exercise compliance are also in line with recent suggestions with regard to the etiological difference between positive and negative affect. Cloninger & Garcia (2014), for example, pointed out evidence (Baker et al., 1992) that suggest that the situational (i.e., that both positive and negative affect are related to the experience of pleasant and unpleasant experiences, respectively; e.g., Warr, Barter & Brownbridge, 1983) and dispositional explanations (i.e., positive affect has it origin in Extraversion, while negative affect arises in Neuroticism; e.g., Costa & McCrae, 1980; Costa & McCrae, 1984) of the origin of positive and negative affect do not to fit the general pattern of data that has been accumulated in support of the independence of positive and negative affect: a primarily situational etiology for positive affect and a primarily dispositional etiology for negative affect (see also Bradburn, 1969; Diener & Larsen, 1984; Emmons & Diener, 1985). This observation led Baker and colleagues (1992) to investigate why positive and negative affect are independent of each other and why they have different patterns of correlation with other variables using data from twins and three-generational families. These researchers found significant negative effects for heritability but not for positive affect. In contrast, positive affect was influenced by shared environmental effects for parents and offspring, assortative mating for spouses, and shared environmental effects for the twin pairs (Baker et al., 1992). Baker and colleagues (1992) concluded that “there may be important (heritable) personality factors that play a critical role in determining levels of negative moods from one person to the next in the family. For positive affect, on the other hand, family resemblance is explained primarily by environmental effects common to family members” (162).

Future studies ought to focus upon the threshold levels of exercise schedules among young and older people aiming at health benefits in different populations. For example, Sénéchal and colleagues (2012) observed that in a study of moderate-severe metabolic syndrome, the severity of disorder and age of patients were determinants of exercise intensity levels. The utility of pursuing exercise programs in aging populations seems, indeed, a growing necessity.

“Exercise to stimulate, not to annihilate. The world wasn’t formed in a day, and neither were we. Set small goals and build upon them.”

Lee Haney

We are grateful to Sophia Garcia for her excellent technical assistance and to Nick Mashkouri, Markus Ekström, and Ulrika Einald for their help with the data collection.

Additional Information and Declarations

Competing Interests

Author Contributions

Human Ethics

Data Deposition

The authors declare there are no competing interests.

Danilo Garcia conceived and designed the experiments, performed the experiments, analyzed the data, prepared figures and/or tables, reviewed drafts of the paper.

Trevor Archer conceived and designed the experiments, performed the experiments, analyzed the data, wrote the paper, prepared figures and/or tables.

The following information was supplied relating to ethical approvals (i.e., approving body and any reference numbers):

After consulting with the university’s Ethical Review Board (University of Gothenburg) and according to the law (2003: 460, § 2) concerning the ethical research involving humans we arrived at the conclusion that the design of the present study (e.g., all participants’ data were anonymous and will not be used for commercial or other non-scientific purposes) required only informed consent from participants.

The following information was supplied regarding the deposition of related data:

Researchgate: https://www.researchgate.net/publication/268807161_Garcia__Archer_2014_Positive_Affect_and_Age_as_Predictors_of_Exercise_Compliance

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
