# Peer review of "Positive affect and age as predictors of exercise compliance"

_PeerJ, doi:10.7717/peerj.694_

## Round 0.1 · original submission · Major Revisions

Dear Authors,Please do the necessary revisions to improve the contents,information on statistics and quality of the presentation of the manuscript before resubmission as it will be re-reviewed by the same two reviewers.

Reviewer 1 ·

Basic reporting

- The article is written in clear English and conforms to professional standards. The introduction is sufficient, fits into the broader field of knowledge and it is relevant to the prior literature.
- Spelling and punctuation mistakes; ilhealth (Line 28), begavior (line 204), ioenergetics (line 558), Archer, 2011, 12 (line 293)
- Missing citation (Line 460-463)
- Wrong space at each new paragraph throughout manuscript
- Some sections of introduction are lengthy or repetitive (e.g., lines 19-20, 24-26)
- Lines 439- 447 do not fit the title Exercise and dietary restriction
- References do not conform to recommended PeerJ style

Experimental design

- No information provided on whether informed that where held participation and under how long time? How data were coded to protect participants' anonymity?

Validity of the findings

- Did you have any missing data? In this case, how this may affect your results?
- Table 1, I think its important to report the possibility of multicollinearity by reporting of Tolerance and VIF value in your data.
- Figure 1: current figure is misleading, it get a picture that only positive Affect contributes to exercises.

Additional comments

- Headline, Exercise and health (line 249). According (WHO) health defined as "a state of complete physical, mental, and social well-being and not merely the absence of disease or infirmity”. I think this whole review is about the health and actually exercises. I think it would have been better if you had started with exercises and health in general in the introduction, and then develop it more in detailed. For example line 276- 279 should be at the beginning of the introduction.

- I suggest that you can have a headline about insulin, diabetes I, II and exercise and then put all the previous finding research about this area in one tittle (It seems there is a lot to say about this field).

- Many diseases that you describe in the introduction are related to age e.g. diabetes II. For more clarity I suggest that you can have a headline on gender, age differences and exercise.
- I suggest that line 65-68 can be moved and merged with line 88-96.
- Do you have any suggestion for future study?

·

Basic reporting

This manuscript contains some important insights however due to the organization, interrupted flow of concepts between sentences (within paragraphs) and repetition, it is difficult for the reader to decipher the messages. It appears to be many papers in one 1. Review of neurobiological effects of exercise; 2. Review of personality traits that predict overeating; 3. Review of the personality traits that predict exercise participation and 4. Review of neurobiological effects of high fat diet and finally the study of affect and exercise behaviour in 280 people.
If the authors intend to review all these then they should do so in a systematic way so the reader is assured of an unbiased representation of the research in these fields;
or
the authors should consider reducing the size of this paper and focusing on the survey. This means reviewing only the aspects important to provide a reasonable introduction on how affect influences exercise participation.

Experimental design

Subjects: how were the subjects recruited? The sample is quite disparate: is there a difference in predictors depending the subgroup (high school, university, workers). were all demographic factors entered into the model; ie age, gender, education? if so then this should be explained/ or why they were excluded.
The authors state there was a 'battery of instruments to measure affect' yet there seems to be only one.

Validity of the findings

see above

---

## Round 0.2 · Major Revisions

Dear Author, please attend to the comments from reviewer 2:

"Unfortunately the authors have not improved the quality of writing in this manuscript and have made almost no revisions to the text. The title suggests that the content will be related to exercise and dietary restriction and some of their common neuroprotective pathways but instead the manuscript covers many other topics such as overeating, depression and diabetes. There is a mixing of concepts between exercise and sedentary behaviour - the two need to be discussed separately since there is a body of literature for both. The manuscript is difficult to follow and topics do not flow logically enough. For example text under the heading ‘Age, gender and exercise’ covers Alzheimer’s disease, diabetes, obesity, Parkinson’s, and neuroinflammation.

There is a massive list of references but it is not clear how this review was conducted-how the search was carried out and how the strength of the evidence was evaluated and if competing findings are being represented."

I think a lot more has to be done to make it academically readable.

Reviewer 1 ·

Basic reporting

No comments

Experimental design

No comments

Validity of the findings

No comments

Additional comments

No comments

·

Basic reporting

Unfortunately the authors have not improved the quality of writing in this manuscript and have made almost no revisions to the text. The title suggests that the content will be related to exercise and dietary restriction and some of their common neuroprotective pathways but instead the manuscript covers many other topics such as overeating, depression and diabetes. There is a mixing of concepts between exercise and sedentary behaviour-the two need to be discussed separately since there is a body of literature for both. The manuscript is difficult to follow and topics do not flow logically enough. For example text under the heading ‘Age, gender and exercise’ covers Alzheimer’s disease, diabetes, obesity, Parkinson’s, and neuroinflammation.
There is a massive list of references but it is not clear how this review was conducted-how the search was carried out and how the strength of the evidence was evaluated and if competing findings are being represented.

Experimental design

see above

Validity of the findings

see above

·

Basic reporting

"No Comments"

Experimental design

"No Comments"

Validity of the findings

"No Comments"

Additional comments

"No Comments"

---

## Round 0.3 · Minor Revisions

Dear Authors,There are minor revisions to be made to your manuscript which I hope your group can do as soon as possible.

·

Basic reporting

The authors have made substantial improvements in the manuscript. There are only minor editorial improvements to make

Experimental design

no comments

Validity of the findings

no comments

Additional comments

Minor editorial improvements
Lines 24-27 second half of the sentence is not clear. Separate into two sentences or rephrase using a semi-colon
Lines 39-40 long sentence that should be split with a period after ‘physical exercise’ in line 39
Line 44. It is not clear what the authors mean by ‘cognitive-affective expressions’. Better to say “its benefits on cognition and affect”.
Lines 48-52-sentence too long and the meaning of the second half is lost. Split into two.
Line 53 What is a ‘signal sensitivity system’? This label appears without context. Please explain what this is and how it relates to your arguments otherwise delete.
Line 61 ‘improvements’
Line 69 Topic moves from cognition to muscle properties with no rationale or introduction provided. Please improve the prose here.
Figure 1 The p is upper case in one disc and lower case in the other

---

## Round 0.4 · accepted · Accept

Dear Authors,Thank you for your manuscript's final submission that has been re-reviewed and accepted. The manuscript will now move to PeerJ's production process.

·

Basic reporting

No comments

Experimental design

No comments

Validity of the findings

No comments

Additional comments

No comments